Aggregation pathways of human γ D crystallin induced by metal ions revealed by time dependent methods

Fernández-Silva Arline 1
French-Pacheco Leidys 1 2
Rivillas-Acevedo Lina lrivillas@uaem.mx 2
Amero Carlos carlosamero@uaem.mx 1
1 Centro de Investigaciones Químicas, IICBA, Universidad Autónoma del Estado de Morelos , Cuernavaca , Morelos , Mexico
2 Centro de Investigación en Dinámica Celular, IICBA, Universidad Autónoma del Estado de Morelos , Cuernavaca , Morelos , Mexico
Posner Mason
Electronic publication date: 2020 Jun 12
Publication date: 2020
Volume: 8
Electronic Location ID: e9178
Received 2020 Feb 12; Accepted 2020 Apr 22
Copyright: ©2020 Fernández-Silva et al.
Copyright year: 2020
Copyright holder: Fernández-Silva et al.
License: This is an open access article distributed under the terms of the Creative Commons Attribution License, which permits unrestricted use, distribution, reproduction and adaptation in any medium and for any purpose provided that it is properly attributed. For attribution, the original author(s), title, publication source (PeerJ) and either DOI or URL of the article must be cited.
License URL: https://creativecommons.org/licenses/by/4.0/

Keywords: Crystallins, Copper, Zinc, Cataracts, Aggregation, Real-time NMR, Human gamma-D crystallin

Funding: CONACYT A1-S-11842 This research was funded by CONACYT A1-S-11842. The funders had no role in study design, data collection and analysis, decision to publish, or preparation of the manuscript.

==============================
Cataract formation is a slow accumulative process due to protein aggregates promoted by different factors over time. Zinc and copper ions have been reported to induce the formation of aggregates opaque to light in the human gamma D crystallin (HγD) in a concentration and temperature dependent manner. In order to gain insight into the mechanism of metal-induced aggregation of HγD under conditions that mimic more closely the slow, accumulative process of the disease, we have studied the non-equilibrium process with the minimal metal dose that triggers HγD aggregation. Using a wide variety of biophysics techniques such as turbidimetry, dynamic light scattering, fluorescence, nuclear magnetic resonance and computational methods, we obtained information on the molecular mechanisms for the formation of aggregates. Zn(II) ions bind to different regions at the protein, probably with similar affinities. This binding induces a small conformational rearrangement within and between domains and aggregates via the formation of metal bridges without any detectable unfolded intermediates. In contrast, Cu(II)-induced aggregation includes a lag time, in which the N-terminal domain partially unfolds while the C-terminal domain and parts of the N-terminal domain remain in a native-like conformation. This partially unfolded intermediate is prone to form the high-molecular weight aggregates. Our results clearly show that different external factors can promote protein aggregation following different pathways.

Introduction

Cataracts are defined as the partial or total loss of transparency of the eye lens due to the formation and deposition of insoluble protein aggregates. Several external factors, such as exposure to UV light, reactive oxygen species, and metal ions, among others, promote the accumulation of damage that causes the destabilization of lens proteins and induce their aggregation over time (Sharma & Santhoshkumar, 2009; Moreau & King, 2012). The average age of cataracts onset is around 50 years and it is one of the leading causes of blindness worldwide, affecting more than 20 million people each year (Lee & Afshari, 2017; Flaxman et al., 2017).

Crystallins constitute more than 90% of the lens proteins and are classified into two superfamilies: the α-crystallins and the βγ-crystallins (Bloemendal et al., 2004). The α-crystallins are members of the family of small heat shock proteins and interact with non-native proteins to prevent their aggregation. The βγ crystallins are structural proteins important to maintain the transparency and high refraction index of the lens (Bloemendal et al., 2004; Andley, 2007). Recently, it has been proposed that the γ-crystallins are able to reduce oxidized protein, forming internal disulfide and trap intermediates, performing an oxidoreductas function (Serebryany et al., 2018).

Human γD crystallin (HγD), one of the proteins in the lens nucleus, consists of 173 residues arranged in two domains, the N and the C-terminal (Kosinski-Collins, 2003; Robinson et al., 2006). Each domain is composed of 8 antiparallel β-strands (A, B, C, D, E, F, G, and H) arranged in two Greek key motifs, held together by a small linker and several hydrophobic interactions at the domain interface (Fig. 1) (Kosinski-Collins, 2003; Basak et al., 2003). Different structural features of HγD have been described to protect and confer stability to the protein; for instance, the tryptophans 42 and 68 in the N-terminal domain, and tryptophans 130 and 156 in the C-terminal domain have been reported as a protection factor against UV radiation (Chen, Callis & King, 2009). Nevertheless, under certain circumstances (changes in pH, temperature or chemical agents), it is possible to unfold the crystallins and promote their aggregation.

Figure 1 HγD structure.

(A) Sequence of HγD crystallin showing the N-terminal (top) and the C-terminal (bottom) domains. Positive-charged residues are shown in blue, negative-charged residues in red, and aromatic residues in yellow. Secondary structure elements are shown. Each strand is named from A to H for each domain and the linker region is shown in purple. (B) Three-dimensional structure of HγD crystallin labeled by strands (PDB id 1HK0). (C) Topology of the N-terminal and C-terminal domain describing the four Greek motifs.

The crystallin aggregation process comprises at least three basic steps. First, the protein sustains some type of damage, which may be due to several different factors. Then, this alteration promotes a conformational change from the native structure to an aggregation-prone structure and, finally, the formation of oligomers and their growth take place. Although the mechanism underlying the aggregation of several crystallins in-vitro has been the subject of numerous investigations over the past decade, the starting process and the pathway remain poorly understood on structural and kinetic grounds (Kosinski-Collins, 2003; Acosta-Sampson & King, 2010).

Several aggregation models have been proposed for the γ-crystallins members. One of the most accepted involves the formation of a partially unfolded intermediate state where the N-terminal domain is completely unfolded while the C-terminal remains folded. This partially unfolded intermediate has been proposed as the starting point for intermolecular interactions between protomers that induce aggregate development (Aguayo-Ortiz et al., 2019; Flaugh, Kosinski-Collins & King, 2005; Whitley et al., 2017). Other models include domain swapping events (Mahler et al., 2011; Moreau & King, 2012), intramolecular disulfide bridge formation (Serebryany & King, 2015), dimerisation (Serebryany et al., 2016; Ray, Hall & Carver, 2016) or a simple condensation mechanism, in which subtle modifications in the protein surface induce association and aggregation (Forsythe et al., 2019; Thorn et al., 2019; Pande et al., 2010; Pande, Mokhor & Pande, 2015; Wong et al., 2019). It has also been reported that the crystallins can form amyloid fibrils (Alperstein et al., 2019). Nevertheless, a detailed molecular characterization of the aggregation pathway has been hampered by the intrinsically dynamics of the process in which these proteins undergo irreversible structural changes and then aggregate over time.

In a previous study, we reported that high concentration of Cu(II) and Zn(II) (up to 10 equivalents) induced the formation of HγD aggregates; moreover, we have found that there is more than one binding site for each of the metal ions (Quintanar et al., 2016). Following studies have continue to investigate the aggregation process due to high concentration of metals (Domínguez-Calva et al., 2018; Ramkumar et al., 2018). In the case of γS-crystallin it was also found that some cations induce the aggregation formation by interacting with cysteine and methonine residues (Roskamp et al., 2019). Nevertheless, it is unlikely that proteins in the lens would be suddenly in contact with a high concentration of metal ions. More likely, cataract formation due to metals in vivo is a slow, accumulative process. In fact, the reported concentration of metals ions in cataract lenses is low, in the order of 10 µg/g (Dawczynski et al., 2002).

In order to gain insight into the mechanism of metal-induced aggregation of HγD under low concentrations of Cu(II) and Zn(II) over time, we evaluated the aggregation process using time dependent techniques. Unlike the approaches introduced before, we did not study the protein with high metal concentration or any other harsh conditions; instead, we studied the minimum damage that was capable to induce aggregation at physiological temperature. We used a combination of several biophysical methods, such as turbidimetry, dynamic light scattering (DLS), fluorescence, nuclear magnetic resonance (NMR), and computational predictions, to obtain information on the molecular mechanisms for the formation of aggregates. We have found that Zn(II) destabilizes the protein and promotes a small structural rearrangement, not detected before, while Cu(II) induces a partial unfolding of the N-terminal domain. We describe the most likely first step into the aggregation process for these two metal ions.

Materials & Methods

All reagents were analytic grade, and used without further purification. CuSO4 and ZnSO4 were used as source of Cu(II) and Zn(II) ions, respectively.

Expression and purification

Recombinant HγD protein (UniProt ID: P07320) was expressed in BL21-RIL Escherichia coli cells transformed with the plasmid pET16b-HγD. The cells were grown in 2XYT medium supplemented with 100 µg/mL ampicillin and 30 µg/mL cloramphenicol, at 200 rpm and 37 °C until an OD600 of 0.9 was reached. Then, HγD protein expression was induced with 1 mM isopropyl D-thiogalactoside (IPTG) overnight at 18 °C. The cells were harvested by centrifugation at 4,000 rpm for 30 min at 4 °C. The pellet was resuspended in 10 mM ammonium acetate buffer pH 7.2, and the cells were lysed by adding 1 mg/mL lysozyme for 30 min and using 10 cycles of 1 min of sonication. The lysed cells were centrifuged at 17,000 rpm for 45 min at 4 °C. The supernatant was applied to a Q-Sepharose column. The fractions containing HγD proteins were determined by SDS-PAGE. The HγD fractions were eluted into a Sephadex 200 column.

Turbidimetry

The effect of Cu(II) or Zn(II) on HγD aggregation kinetics was followed by turbidimetry. Electronic absorption data at 405 nm were acquired on an Agilent 8453 UV-Visible diode array spectrophotometer. HγD protein samples were 50 µM in 10 mM amonium acetate pH 7.2 and 50 mM NaCl. For the turbidity assays HγD was incubated in the presence and absence of 1.5 equivalents of CuSO4 or ZnSO4 at 37 °C for 12 h. The turbidity changes were measured at 405 nm every 60 s during 12 h. The variation of the measurement were estimated from three successive recorded data points. Experiments were performed in duplicates for each condition. A linear regression was applied to the linear phase of the plot of turbidity versus time in order to calculate the slope.

Dynamic Light Scatering (DLS)

The changes in the size of the protein or its aggregates were measured by DLS. Measurements were performed on a Malvern Zetasizer Nano ZSP spectrophotometer, with a scatter angle of 173°. HγD samples were 50 µM in 10 mM amonium acetate pH 7.2 and 50 mM NaCl. All samples were centrifuged for 5 min at 9,000 rpm and filtered through a 0.22 µm syringe filter before measurements. The changes in size over time were measured by incubating HγD in the absence and presence of 1.5 equivalents of CuSO4 or ZnSO4 at 37 °C every 1.6 min during 12 h. Three runs with 10 scans of 10 s were obtained for each measured data point, then 3 sequential points were used to calculate the variation in the measurements. Data were analyzed by the cumulant or the distribution methods implemented in the SEDFIT software (Schuck, 2000). Experiments were performed in duplicates for each condition.

The data were used to obtain translational diffusion coefficients through measurement of the correlation coefficient. The hydrodynamic radius, RH, was obtained from the diffusion coefficient, D, via the Stokes-Einstein equation: RH=KBT∕6πηD

where KB is Boltzmann’s constant, T is the temperature, and η is the viscosity of the solution. We used 0.08872 poise for the viscosity. Prediction of hydrodynamic radius from the structure were calculated using HYDROPRO software (Ortega, Amorós & García de la Torre, 2011). The coordinates in PDB 1HK0 were used as input for a monomer protein. Values of partial specific volume of 0.7 cm3/g were used.

Fluorescence

Conformational changes of HγD protein in the presence of Cu(II) or Zn(II) were followed by fluorescence. Intrinsic fluorescence spectra were recorded at 37 °C using a fluorescence spectrophotometer Cary-Eclipser (Agilent). The emission spectrum was recorded in the range of 300 and 500 nm using an excitation wavelength of 295 nm. 50 µM HγD in 10 mM of amonium acetate and 50 mM of NaCl pH 7.2, were incubated at 37 °C in the presence and absence of 1.5 equivalents of CuSO4 or ZnSO4. The emission spectra were collected every minute during 12 h.

The emission signals at 325 and 350 nm were used to calculate the 350/325 ratio. This ratio was used to estimate the change in the Trp environment of the protein (normalized fraction) based on the protein temperature unfolding, where the baseline corresponds to 0% unfolded and the maximum value of 350/325 ratio (∼1.44) to 100% unfolded.

NMR Spectroscopy

All NMR spectra were recorded on a Varian 700 MHz VNMR-S spectrometer equipped with a cryogenically cooled triple resonance pulsed field gradient probe. Backbone resonance assignments at 37 °C were transfered from previously reported values at 20 °C (Quintanar et al., 2016). All spectra were processed with NMRPipe (Delaglio et al., 1995) and analyzed using CARA (Keller, 2004). 100 µM HγD crystallin in 10 mM ammonium acetate pH 7.2 and 50 mM NaCl samples were prepared.

A series of two dimensional 1H-15N Heteronuclear Single Quantum Coherence (HSQC) experiments were collected every 30 min during 12 h in the absence and presence of 1.5 equivalents of CuSO4 or ZnSO4 at 20 °C and 37 °C. Signal intensities and position were measured for each spectrum and analyzed independently. The intensities were normalized individually with respect to the first data point, monitored as a function of time, and then fit to a first order exponential equation: It= exp−at

where I(t) is the peak intensity at time t, and a is the mean-time constant of the process. Error estimates of the fit parameters were obtained by Monte Carlo simulations of 500 iterations, using the noise as an estimate of the error for each measurement.

Weighted average chemical shift perturbations (CSP) were calculated from: ΔCSP= |Δδ1H+1∕a|Δδ15N|

where Δδ (1H) and |Δδ (15N) are the differences in chemical shift between different HSQC spectra, with a=8 for Gly, and 6 otherwise.

Computational methods

The MIB server (Metal Ion-Binding Site Prediction; Lin et al., 2016) was used to predict potential metal binding sites. The structure from the PDB (entry 1HK0) was used with the default settings. Molecular docking was accomplished using a driven flexible docking protocol in the High Ambiguity Driven biomolecular DOCKing (HADDOCK) software (Van Zundert et al., 2016). The binding residues for the interacting models between HγD monomer and metal ions were selected using the NMR chemical shift perturbation data and the MIB prediction. The lower energy score of HADDOCK was used to select the best structures. Those are based on a weighted sum of electrostatics, desolvation and Van der Waals energy terms, along with the energetic contribution of the restraints used to drive the docking.

Results

The cataract formation process in-vivo is a slow, accumulative, low dose process, most likely due to multiple factors. One of these factors could be the exposure to metal ions. It has been reported that Cu(II) and Zn(II) induce HγD crystallin aggregation in a metal-concentration dependent manner (up to 10 equivalents) and temperature (from 37 °C to 60 °C) (Quintanar et al., 2016). To provide a scenery closer to the actual physiological process, we decided to study the mechanism of aggregation induced by low metal ion doses as a function of time at 37 °C. Taking into account that there are probably multiple binding sites, and that there is no significant aggregation with 1 equivalent or sub-stoichiometric concetrations of the metal ions, 1.5 equivalents of Cu(II) and Zn(II) were selected as the minimum concentration to observe aggregation at 37 °C for 12 hr.

Effect of Cu(II) and Zn(II) in HγD aggregation over time

To asses the formation of aggregates opaque to light, like the ones found in cataracts, we measured the changes in absorbance at 405 nm at 37 °C for 12 hr. In the absence of any metal ion, HγD crystallin shows a constant optical density (OD) and remains soluble during the experiment time, confirming that the free native protein does not change its light scattering properties under this condition. However, in the presence of either Zn(II) or Cu(II), there was a large increase in turbidity, indicating the formation of translucent states of HγD in solution as a function of time (Fig. 2).

Figure 2 Effect of Zn(II) and Cu(II) as reported by turbidity assays at 37 °C.

(A) Absorbance at 405 nm as function of time of HγD crystallin in the absence (black line) and presence of 1.5 equivalents of Zn(II) (red). (B) HγD crystallin in the absence (black line) and presence of 1.5 equivalents of Cu(II) (blue). The lag phase for copper is shown. The change in absorbance is due to the formation of aggregates that scatter the light.

The addition of Zn(II) induces an immediate steady increase in turbidity, with rate of 11.67 × 10−4 ± 1.2 × 10−5 per min in the first 280 min (Fig. 2A), followed by a decrease of the OD observed over longer times (Fig. S1A). This decrease is the result of the formation of high molecular weight aggregates that precipitate. In contrast to that found with the addition of zinc, the addition of Cu(II) displays first a plateau of about 50 min, followed by a rapid increase phase with rate of 13.7 × 10−4 ± 5.5 × 10−5 per min in the next 80 min (Fig. 2B). After that, there is also a decrease in the rate at longer times (>130 min), which corresponds to the visual appearance of precipitations in the cuvette (Fig. 2B and S1B). Although both divalent metal ions induced an increase in the turbidity, the kinetics and the final states are significantly different, with copper presenting a bi-phase behavior, indicative of different aggregation mechanisms.

Interestingly, the soluble protein at the end of the experiments remains high, with 80 to 90% of the original concentration as measure by UV spectroscopy. This shows that most of the light scattering is due to a low fraction of protein forming large aggregates, and this process in principle could continue for longer times.

Formation of high molecular weight, light-scattering aggregates

In order to get insight on the sizes of HγD aggregates induced by the metal ion interactions over time, samples were analyzed by dynamic light scattering (DLS). As a reference, we calculated the predicted hydrodynamic translational diffusion coefficient predicted by HYDROPRO (Ortega, Amorós & García de la Torre, 2011) software from the monomer and a hypothetical dimer using the crystallographic coordinates (PDB entry 1HK0). The obtained radius and translation diffusion coefficients were RH = 2.19 nm and D = 1.122 × 10−7 cm2/s for the monomer, and RH = 3 nm and D = 0.81 × 10−7 cm2/s for the dimer.

Unexpectedly, DLS measurements of free HγD show a correlation curve consistent with a polydisperse solution, with at least two different species. The most abundant species, that accounts for ∼80% of the total signal, has an apparent diffusion coefficient of D = 1.06 × 10−7 cm2/s which corresponds to an apparent RH of 2.3 nm (Fig. S2). The second species has an apparent radius around ∼75 nm, and corresponds to 16% of the total signal. It is worth mentioning that DLS measurements are significantly more sensitive to larger molecules; therefore, in this case the percentage of the detected signal is an under estimation of the populations. The results suggest that, under these conditions, HγD consists mostly of monomeric protein, however, there are larger oligomeric species also present at the start of the experiments. After the incubation of free HγD at 37 °C for 12 h, we continue to detect two species, but the percentage of total signal changes to 57% for the monomer and 41.5% for the oligomers (Fig. S2). Interestingly, the sample remains completely transparent judging by the turbidity and visual inspection. All the remaining experiments were performed with samples that contain at least 80% of the signal from the monomeric protein, and analyzed taking in consideration two main species: small (monomeric) and large (oligomer).

Upon addition of 1.5 equivalents of Zn(II) ions, the correlation curves shift to the right, consistent with a size increase due to oligomerization induced by metal-binding (Fig. 3A). The monomer signal percentage changes from 84.7% for free HγD to less than 10% in the first data point after the addition of the metal ion, while the signal from the oligomeric species goes from 11.6% to more than 88%. The calculated hydrodynamic radius over time yielded a global steady increase up to 800 nm before 70 min, with a rate of 12.4 ± 0.16 per min (Figs. S3 and S4). This rate matches well the turbidimetry measurements rate, confirming that the increase in size of the oligomers is in fact responsible for the solution turbidity (Fig. S3C). At longer times (>80 min), the data shows dispersion, most likely due to the precipitation of larger aggregates and polydispersity of the sample (Fig. S3).

Figure 3 Protein oligomerization induced by metal-binding.

(A) Correlation coefficient of HγD in the absence (grey line) and presence of 1.5 equivalents of Zn(II) at 1.6 min (red), 14.4 min and 24 min after the addition of the metal ion. (B) Correlation coefficient of HγD in the absence (grey line) and presence of 1.5 equivalents of Cu(II) (blue) at 1.6 min, 14.4 min, and 24 min after the addition of the metal ion. Measurements of different times yielded a shift to the right indicating a size increase due to oligomerization. From the shape it can be seen that more than one species form part of the population sample. (C) Estimated hydrodynamic radius separated by the two species observe in the presence of Cu(II) as a function of time: small (monomeric) species (right) and the large species (left) (blue circles—right axes). The grey bars indicate the signal percentage of the species changes over time during the protein oligomerization kinetics (left axes). The predicted RH for HγD monomer and dimer calculated using HydroPRO are shown. All samples were centrifuged and filtered before measurements.

The effect of 1.5 equivalents of Cu(II) ions was different, corroborating a distinctive aggregation mechanism. Immediately after copper addition, the monomeric species retains a similar size with similar signal percentage as shown in Figs. 3B and 3C. Interestingly, the subsequent measurements yield a small increase in the RH from 2.2 to ∼2.7 nm while the percentage of total signal gradually decreases until it reaches less than 10% at ∼100 min (Fig. 3C). On the other hand, the signal corresponding to the oligomer species shows the opposite trend, with a gradual increase reaching more than 90% of the total signal at 100 min and sizes up to 500 nm (Fig. 3C). At longer times (>300 min) there was dispersion in the data due to precipitation as observed by turbidimetry (Fig. S3). Therefore, the lag phase observed in turbidimetry corresponds to the transition between mostly small molecule species to the formation of the large aggregates. Results shown at Table 1.

Overall, these data confirm that both Cu(II) and Zn(II) induce the formation of high molecular weight aggregates opaque to light with different mechanisms.

Conformational changes in HγD in the presence of Cu(II) and Zn(II)

Fluorescence spectroscopy was used to detect changes in the local environment of the tryptophan residues. The maximum emission for the intrinsic fluorescence of the native protein is around 325 nm, whereas the maximum is shifted to 350 nm when the protein is completely unfolded. Therefore, the effect of the metal ions in the HγD folding state can be evaluated by following the ratio 350/325 over time. In the absence of metal ions, HγD maximum emission remained at 325 nm during the 12 h of analysis (Fig. S5) and the 350/325 ratio does not change, indicating that the protein remains folded throughout the experiment (Fig. 4).

Table 1 Hydrodynamic radius.

RH from different species obtained by DLS analysis at two time points.

	Apparent radius—Fraction of total signal	
HγD+	First data point	After 24 min	
Zn(II)	Monomer: ∼2.5 nm–9.7%	Monomer: ∼2.42–0.98%	
Oligomers: ∼93.4 nm–88.7%	Oligomers: ∼319.3 nm–89.7%	
Cu(II)	Monomer: ∼2.27 nm–90%	Monomer: ∼2.7 nm–38%	
Oligomers: ∼49.04 nm–5.8%	Oligomers: ∼75.6 nm–58.3%	

Figure 4 Changes of the Trp environment induced by metal ions.

(A) Normalized 350/325 fraction in the absence (black) and presence of 1.5 equivalents of Zn(II) (red) as a function of time. (B) Normalized 350/325 fraction in the absence (black) and presence of 1.5 equivalents of Cu(II) (blue) as a function of time. This ratio was used to estimate the percentage of change in Trp environment of the protein (normalized fraction) based on protein temperature unfolding experiments, where the baseline corresponds to folded protein and the maximum value of 350/325 ratio (∼1.44) correspond to 100% unfolded proteins.

In the presence of Zn(II), we observed a steady and slow change in the 350/325 ratio over time, going from 0.63 to 0.68, suggesting a small change in the Trp environment during the aggregation process (Fig. 4A and S5). This result is in concordance with previous reports that indicate that after the interaction of HγD with Zn(II), the secondary structure signal changes 10% (Quintanar et al., 2016).

In contrast, HγD crystallin in presence of copper presents a bi-phase time course, with a rapid initial phase followed by a slower steady change (Fig. 4). Interestingly, the transition from the rapid phase to steady state coincide with the lag phase of the copper-induced aggregation observed in turbidity and DLS. The emission maximum shift to 330 nm after 12 h of incubation at 37 °C (Fig. S5). This suggests that copper induces a structural rearrangement that is not observed for zinc binding.

Residue specific binding and aggregation mechanism

Due to its atomic resolution capabilities, it was desirable to follow the process by NMR, to detect residue specific changes produced by the binding and aggregation induced by Zn(II) or Cu(II). The signals for each residue were analyzed by the changes in peak position or intensity. It has already been reported, the interaction of HγD with metal ions produces several changes in the chemical shifts at 20 °C, suggesting several possible binding sites without oligomerization (Quintanar et al., 2016) and since the addition of Zn(II) to HγD induces immediate HγD oligomerization at 37 °C, we proceeded to analyze in detail the chemical shift perturbations (CSP) differences between 20 °C and 37 °C.

After the addition of Zn(II), we can group all the CSP in 4 different regions for both temperatures (Fig. 5). Region 1 contains His 22, which has been implicated in HγD oligomerization induced by Zn(II) (Domínguez-Calva et al., 2018). The residues in this region suffer small CSP, even at sub-stoichiometric concentrations of Zn(II), most likely reflecting metal binding and oligomerization.

Figure 5 Zn(II) binding to HγD crystallin as detected by NMR.

(A) HγD chemical shift changes induced by the binding to Zn(II) at 37 °C mapped onto the protein structure (1HK0). Different regions are color-coded with a linear gradient from light (small CSP changes) to dark (largest CSP changes): region 1 (blue), region 2 (red), region 3 (purple), and region 4 (green). Unassigned residues are shown in dark gray. Residues H22, W42, W68, W130 and W156 are shown as sticks (B) HγD Chemical Shift Perturbation (CSP) after the addition of Zn(II) at 20 °C and 37 °C. Bars have been color-coded based on affected regions as described above. Cyan asterisks show buried residues that suffer significant changes. Sidechain CSP for W42, W68 and W130 are shown at the far right. Possible binding sites are shown in Fig. S11.

Region 2, at the N-terminal domain, contains residues exposed to the solvent that can potentially bind Zn(II), but also contains residues completely buried at the core of the protein or at the interface between domains. Some of the buried residues present different CSP at different temperatures, for instance Leu 45, Leu 53 and the side chain of Trp 42 only shift significantly at 37 °C. Therefore, some of the detected changes in this region could be reporting a potential binding to Zn(II), but other changes are reporting a small structural rearrangement at the core of the N-terminal domain and in the interface between domains, most likely required for oligomerization.

Region 3 comprises the linker between domains and the top of the C-terminal domain. It also contains some buried residues at the core or at the domain interface. This region is sensitive to small quantities of metal ion, and, in some cases, we observe two sets of peaks, representing Zn(II) binding and a small rearrangement. Finally, region 4 contains exposed residues in the opposite extreme of the C-terminal domain, related to another metal binding site. This site presents a significant increase in perturbation at 37 °C.

Overall, the chemical shift perturbation for Zn(II) at 37 °C shows changes at similar regions to those recorded at 20 °C, however, at 37 °C there are more residues affected or with larger perturbations (Fig. 5). Based on the absence of aggregation at 20 °C, the specific differences with 37 °C are likely due to the first steps in the aggregation process. Regions 2 and 3 present an increase in the number of affected buried residues with temperature, suggesting that the rearrangements in the core and at the inter-domain interface, enhanced by the raise in temperature, is required for the oligomerization. It should be noted that the spectra do not contain new signals that could correspond to an unfolded event. All the data suggest that the binding to Zn(II) at 37 °C induces a subtle but detectable small conformational rearrangement without unfolding of the protein during the aggregation process. Residues for each region are shown at Table 2.

Table 2 Zn(II) regions.

Different chemical shift perturbation regions induced by the binding to Zn(II) at 37 °C.

Region 1	Region 2	Region 3	Region 4	
Y16, E17, S20, D21, H22, and Y28	I3, D38, S39, G40, C41, W42, M43, Y45, E46, L53, F56, L57, R58, R59, G60, D61, Y62, S74, and R79	H83, S84, S86, H87, R88, I89, E95, M101, E103, T105, D107, V125, L126, E127, G128, W130, L132, L144, L145, G148, W156, T159, R162, G164, R168, I170, D171, and F172	L111, Q112, D113, R114, and F115	

As Cu(II) is a paramagnetic ion, residues in close proximity to the metal ion are no detectable by NMR. Hence, the metal binding was estimated by following the changes in intensity, which correlate with the distance to the metal ion. The differences in the HγD spectra due to the metal ion interaction at 37 °C is very similar to the changes at 20 °C (Fig. S6). Therefore, the measured changes for copper only reflect the binding to the metal ion. Using the signal changes, it is possible to accommodate at least two Cu(II) at residues 80-90 and 110–120 as the most likely binding sites at 37 °C, similar to what has been reported before at 20 °C (Quintanar et al., 2016).

Real time NMR reports HγD conformational changes and unfolding

To detect residue specific changes produced by the aggregation process over time, a series of two dimensional HSQC NMR spectra were recorded every 30 min for 12 hr at 37 °C. The spectra were analyzed to evaluate changes in the peak position and intensity for each residue as a function of time. The HγD spectra without metal ions did not present changes in chemical shift over 12 h at 37 °C, indicating that the protein does not have any structural changes under these conditions during this period of time (Fig. S7).

Real-time NMR in the presence of Zn(II) at 20 °C shows no noticeable spectra changes during the experiments, in concordance to the lack of any visual turbidity in the sample. The experiments performed at 37 °C show a general decrease in intensities, probably due to the steady formation of large oligomeric species invisible for NMR (Fig. S8). The average decrease in the intensities was 0.7 after 12 h, and it was observed in both domains, without a clear specifically affected region of the protein (Fig. S8). These results support the notion that the more relevant process to induce oligomerization is faster than the NMR experimental scale. Therefore, when the first spectrum is completed, the changes observed in the NMR spectrum already contain information for the metal binding as well as any necessary changes to induce aggregation.

On the other hand, the copper-induced aggregation process presents a complex process, with several signals changing intensities and shifting over time (Fig. S9). Moreover, some residues present double peaks, indicating conformational fluctuations in the structural ensemble. To simplify the analysis, we grouped residues with similar characteristics. The intensity decay meantime (rate−1) for each group was estimated by fitting all the residue data in the same group, to a single exponential decay equation.

The first group consists of 51 residues whose signal intensities or positions do not change substantially during the experiment (Fig. S9A). These resonances originate from regions that have native-like chemical shifts, for both the amide proton and the amide nitrogen nucleus. Therefore, they identify regions in which a native environment remains unaltered in the aggregation process. Interestingly, most of these residues, 42, are located in the C-terminal domain, while the remaining 9 form a patch in one side of the N-terminal domain (strands DN, FN and GN) (Fig. 6B and S10).

Figure 6 HγD conformational changes induced by metal ions followed by real time NMR.

(A) Intensity mean time changes mapped onto the crystal structure of HγD crystallin (1HK0) and color-coded based on the profiles with a linear gradient ramp from dark blue (no changes) to red (small mean time), for (A) Zn(II) and (B) Cu(II). Unassigned residues are shown in dark gray. (C) Intensity mean-time changes (I/Io profiles) for the HSQC signals over time for each residue with 1.5 equivalents of Cu(II).

The second group consists of cross-peaks of backbone amides of residues with a small decrease in their intensities (Fig. S9B). A total of 24 signals are in this group, mainly in regions in close proximity to the first group. The third group comprises 20 resonances, which present a moderate intensity reduction (Fig. S9B). These residues report structural changes, but without going to an unfolded state. The last group consists of 31 peaks where the intensity of the well-dispersed cross-peaks substantially decreases over time, and native-like resonance peaks almost disappear (Fig. S9C). All of these signals are located at one side of the N-terminal domain (strand BN, HN and EN) (Figs. 6B, 6C and S10). Notably, there are also some new resonance peaks which appear in the unfolding region and whose intensities increase over time with a complementary trend to the signals in group four. These results strongly suggest that the residues in this region are residues that go through an unfolding event, and result in the new unfolded signals. Residues for each group are shown at Table 3.

Table 3 Cu(II) groups.

Different conformational groups induced by Cu(II) ions detected by real-time NMR. Group 1 comprise ∼30% of the residues of the protein, group 2 (∼14%), group 3 (∼12%), and group 4 (∼18%). The remains residues were not assigned or overlapped.

Group 1 (30%)	Group 2 (14%)	Group 3 (12%)	Group 4 (18%)	
Q12, G13, D38, R58, Y62, A63, M69, S72, S74, R88, R90, L91, E93, R94, E95, D96, Y97, R98, G99, Q100, M101, I102, E103, F104, T105, D107, E119, S122, L123, N124, V125, E134, L135, S136, N137, Y138, R139, R152, Q154, W156, G157, A158, T159, N160, A161, V163, G164, L166, R167, V169, and I170	R9, G10, C32, V37, Y55, D64, G70, I89, C108, D113, F115, V131, Y133, G140, R141, Q142, Y143, L145, M146, G148, D149, Y150, R151, and S165	Y6, D8, F11, Y16, A35, R36, E46, Q47, S51, Q54, G60, H65, Q67, W68, E106, H121, E127, G128, D155, and R162	I3, L5, E7, R14, E17, C18, S20, H22, L25, Y28, L29, S30, R31, N33, S34, S39, M43, L44, Y45, N49, Y50, G52, F56, L57, V75, R76, S77, C78, R79, L80, and I81	

Figure 6 shows the intensity decay meantime value mapped onto the HγD structure, and it is clear that the most drastic changes occurred at strands AN, BN, CN, EN and HN. These results, along with those observed by fluorescence (Fig. 4B), indicate that Cu(II) induces a partial unfolding of the protein N-terminal domain.

Computational methods

In order to estimate possible Zn(II) binding sites, the NMR data were compared to the predicted binding sites by the MIB server. MIB searches for sequence similarities between the target protein and proteins with reported metal binding sites . From the selected possible residues, a docking protocol was used to define the binding site at atomic level using HADDOCK and the experimental data. We selected the best structures to describe the metal ion binding sites according to the lowest energy score. From the results we can select four possible binding sites: Residues S20, D21, H22 in region 1; G40, C41, I81, H83, D171 at region 2 and 3; and D113 at region 4 (Fig. S11). Each site involves main chain deprotonated amides and the sidechains of His, Ser, and Asp.

In a similar manner, the NMR data and the MIB results were used to produce docking models to predict HγD Cu(II) binding sites (Fig. S12). The best models were Cu(II) binding to S87, H88 and D113.

Discussion

Cataracts are a slow accumulative process, in which small, local damage caused by external factors induces protein aggregation over time. Metal ion interaction with crystallins is one of the risk factors involved in the development of this pathology. In order to understand the first steps of aggregation under conditions that mimic the slow accumulative process of the disease, we decided to study the non-equilibrium process with the minimal metal dose that triggers HγD aggregation.

Our results show that 1.5 equivalents of Cu(II) or Zn(II) induce opacification of the HγD solution over time at 37 °C, as observed by turbidimetry, whereas the free protein solution remains clear in the same period of time (Fig. 2). Meanwhile, the DLS experiments show the formation of large assembles of protein in the presence of both metal ions (Fig. 3). The measured rates for both techniques correlated well, indicating that indeed the opacification was due to the formation of large aggregates (Fig. S3C). At longer times, there was a decrease in turbidity and size, due to protein precipitation. Interestingly, measurement of the soluble fraction after the experiments, shown than between 70 and 80% of the proteins remains soluble, indicating that the opacification observed was due to a small percentage of protein forming large aggregates.

Even though both metal ions induce the formation of high molecular weight light-scattering aggregates, the aggregation pathway by each one was different. At the first measurement after the addition of Zn(II), the hydrodynamic radius of HγD mostly corresponds to a large oligomer as shown by DLS (Fig. 3). Hence the HγD aggregation induced by Zn(II) results in the rapid formation of soluble oligomers that continue to increase linearly until about 400 nm at 80 min.

The conformational changes experienced by the protein during the aggregation process were followed by fluorescence. In contrast to turbidimety and DLS, in which most of the signal comes from the larger sizes species, in fluorescence, there is no dependence on the particle sizes, and therefore we were able to measure the signal that includes small and large molecules at the same time. The measurements as a function of time show that the presence of Zn(II) induced a small change in the tryptophan chemical environment as detected by the 350/325 ratio, suggesting a subtle conformational rearrangement as the oligomers are forming.

Indeed, the real-time NMR data show a subtle and global intensity decay of the Zn(II)-HγD complex over time, consistent with the slow formation of larger aggregates. The spectra do not present any new signal that could represent unfolded species or species with a significant conformational change. These results reinforce the notion that the most relevant process that induces oligomerization due to zinc is faster than the NMR experimental time scale. Therefore, when the first spectrum at 37 °C is completed, the changes observed in the spectrum already contain information for the metal binding as well as any necessary change to induce the aggregation. The fact that the Zn(II)-HγD complex at 20 °C does not aggregate after 12 h, suggests that the difference between the perturbed signals observed at the first point in the spectra at 20 °C and 37 °C would allow us to dissect the changes that induce the protein oligomerization.

With 1.5 equivalents of Zn(II), we observed CSP around four regions (Table 2). These regions include His 22, Asp 38, Asp 61, Cys 111, and Asp 113, which were also predicted by the MIB server as possible binding sites, and also His 83 and His 87. It is worth mentioning that during titration experiments there was not a clear saturation site filled before the other possible sites, as already reported (Quintanar et al., 2016). This and the fact that just one equivalent of Zn(II) does not induce aggregation, strongly suggests that there is not a preferential binding site, but most likely there are several different interaction sites, and the overall effect is the sum of all small contributions.

Interestingly, we observed several differences in the CSP between 20 °C and 37 °C, some of them located in buried residues, for instance W42, E46, L53, and L145 at the core of the domains, and C41, M43, Y45, L144 and R168 at the interface between the N and C-terminal domains. Due to their location, these changes cannot be due to direct binding to Zn(II) in the native structure; instead, they may be attributed to a small conformation rearrangement between the N and C-terminal domains. Recently, residue Leu 53 has been implicated as an important residue in an all atom-simulation of the oligomerization process for a HγD mutant. In this simulation, the protein undergoes a change in the relative domain orientation, without an unfolding event, and this rearrangement promoted the aggregation (Wong et al., 2019).

We also detected changes in the Trp 42, and Trp 68 sidechain chemical shifts, which corroborates the subtle changes observed by fluorescence.

The HγD aggregation induced by Zn(II) has been demonstrated to be depended on His 22 at region 1 and be reversed by EDTA, suggesting the formation of metal bridges between HgD monomers (Domínguez-Calva et al., 2018). Therefore, region 1 must function as an anchor point in the aggregation process.

Putting all this together, the results indicate that HγD has several Zn(II) binding sites, most likely at regions 1, 2-3 and 4, probably with similar affinities. At region 1, the Zn(II) binds to the surroundings of His 22, that works as an anchor to mediate the bridge formation with residues of regions 2–3 and 4. Either during bridge formation or in order to form the bridge, the protein has to suffer small conformational changes within and between domains.

On the other hand, the HγD aggregation kinetics in the presence of Cu(II) followed by turbidimetry and DLS show a sigmoid behavior (Fig. 2). This behavior indicates that there are at least two processes occurring during the aggregation. The first DLS data point after the addition of metal shows a correlation function similar to the free protein, where the main population that we can deconvolute corresponded to the monomer. In the next measurements, we detect a small increase in hydrodynamic radius going from 2.4 to 2.7 nm. Interestingly, the theoretically calculated hydrodynamic radius based on the structure was 2.19 for a monomer and 3 nm for a dimer. Therefore, it seems possible that the measured population with radius of 2.7 nm corresponds to a partially unfolded intermediate. After 80 min, this small population generates less than 10% of the signal, and we only observe large oligomer species (Fig. 3).

The changes in the Trp environment measured by the fluorescence emission 350/325 ratio, suggest that Cu(II) binding to HγD induces a partial unfolding of the protein (Fig. 4), that could represent the first process of the aggregation kinetics.

In an attempt to better understand the details of the HγD unfolding and aggregation events in the presence of Cu(II), we used NMR real-time techniques. The NMR experiments report small to moderate size particles, whereas the large oligomers formed will become NMR silent. Therefore we are just observing the first changes in conformation and the beginning of the aggregation. The data show a complex process with changes corresponding to fast and slow intermediate exchange. The HγD NMR spectra in the presence of Cu(II) show several residues in the N and C-terminal domains that remain unchanged during all the experiment. Nevertheless, other residues, mostly at the N-terminal domain, were substantially affected after the interaction with the metal ion; even more, there are a few signals emerging in the unfolded region of the spectra. These results are consistent with a partially unfolded N-terminal domain in the presence of Cu(II).

The complete unfolding of the N-terminal domain has been proposed as general intermediate on the HγD aggregation pathway. However, our measurements indicated that after Cu(II) binding, at different sites, the first step is the partial unfolding of the N-terminal domain, specifically strands AN, BN, CN, EN and HN, while most of strand DN, FN and GN keep their native-like conformation (Fig. 6 and S10).

Even-though, the complexity of the metal induces aggregation, which involve binding, specific conformational changes and oligomerization, prevents us from completely dissect the binding sites for copper, the data permit us to group together residues with similar behavior.

A recent study has suggested that Cys 111 was implicated in copper-induced oxidation of crystallin as the mechanism of aggregation (Ramkumar et al., 2018). Contrary to our studies, the authors did not detect aggregation at low concentrations on short times and hence performed the experiments with higher copper equivalents (from 3 to 5). In our CSP copper data the region 110–120 emerges as a possible binding site. Nevertheless, we do not observe changes during the aggregation process measured by real-time NMR techniques. Therefore, while Cys 111 might be the major player of copper-induced aggregation at high Cu(II) equivalents, it does not seem to be the determining event in a slow accumulative aggregation process.

Conclusions

The measurement of dynamic changes in the human gamma D crystallin due to the minimum dose of metal ions that is required to promote aggregation at physiological temperature over time has provided us with important insights both into the first steps of the aggregation process and a more accurate picture of the global pathways.

Our experimental data show that Zn(II) and Cu(II) interact in multiple sites with HγD and induce the formation of large aggregates opaque to light over time. Comparison of the data for free and zinc-bound HγD at different temperatures reveals structural changes necessary to induce the formation of aggregates by a metal bridge formation, while the site-specific measurements of the effect of copper-bound HγD as a function of time, highlight a partial unfolding of the N-terminal domain which makes the protein more prone to aggregation. Remarkably, even-though the zinc does not induce protein unfolding, it still forms large aggregates rapidly.

We can conclude that each metal ion induces aggregation following different pathways, underscoring the premise that cataract formation is a more complex process than generally assumed, that can process with or without protein unfolding. An accurate descriptor of the process will need to incorporate different contributions for different factors.

Supplemental Information

Supplemental Information 1 Supplemental Figures

Figure S1: Effect of Zn(II) and Cu(II) divalent metal ions as reported by turbidity assays. Figure S2: DLS measurement of HγD at 37 °C at different times. Figure S3: Effect of Zn and Cu divalent metal ions on the hydrodynamic radius. Figure S4: Different replicas of protein oligomerization induced by metal-binding. Figure S5: Fluorescence spectrum of HγD at diferents time in absense and presence of metals ion. Figure S6: Cu(II) binding to HγD crystallin as detected by NMR. Figure S7: HSQC spectra of HγD crystallin at different times. Figure S8: HγD conformational changes induced by metal ion followed by real time NMR. Figure S9: Intensity decay for selected residues followed by real time NMR. Figure S10: Scheme of unfolding region of HγD crystallins in presence of 1.5 equivalents of Cu (II) by NMR. Figure S11: Potentially binding sites of Zn (II) obtained by MIB and NMR. Figure S12: Potentially binding sites of Cu (II) obtained by MIB and NMR.

Click here for additional data file.

Supplemental Information 2 DLS raw data

Z(ave) as function of time of HγD in the absence and presence of metals

Click here for additional data file.

Supplemental Information 3 Raw turbidimetry data

Absorbance at 405 nm as function of time of HγD in the absence and presence of metals

Click here for additional data file.

Supplemental Information 4 Fluorescence

Fluorescence 350/325 as function of time of HγD in the absence and presence of metals

Click here for additional data file.

Supplemental Information 5 RT-NMR raw data: HgD - Cu 1

Click here for additional data file.

Supplemental Information 6 RT-NMR raw data: HgD + Cu 2

Click here for additional data file.

Supplemental Information 7 RT-NMR raw data: Zn 1

Click here for additional data file.

Supplemental Information 8 RT-NMR raw data: Zn 2

Click here for additional data file.

The authors acknowledge LANEM for NMR instrumentation. Part of the research was performed at the LabDP–UAEM. The authors thank Nina Pastor for for stimulating discussions. Arline Fernández-Silva and Leidys French-Pacheco are thankful to CONACYT.

Additional Information and Declarations

Competing Interests

Author Contributions

Data Availability

The authors declare there are no competing interests.

Arline Fernández-Silva conceived and designed the experiments, performed the experiments, analyzed the data, prepared figures and/or tables, authored or reviewed drafts of the paper, and approved the final draft.

Leidys French-Pacheco conceived and designed the experiments, analyzed the data, authored or reviewed drafts of the paper, and approved the final draft.

Lina Rivillas-Acevedo conceived and designed the experiments, performed the experiments, analyzed the data, authored or reviewed drafts of the paper, and approved the final draft.

Carlos Amero conceived and designed the experiments, performed the experiments, analyzed the data, prepared figures and/or tables, authored or reviewed drafts of the paper, and approved the final draft.

The following information was supplied regarding data availability:

The raw data are available in the Supplemental Files.

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
