# Peer review of "Aggregation pathways of human γ D crystallin induced by metal ions revealed by time dependent methods"

_PeerJ, doi:10.7717/peerj.9178_

## Round 0.1 · original submission · Minor Revisions

Thank you for submitting your manuscript to PeerJ. Based on the very positive comments of two reviewers I invite you to resubmit after making minor revisions. Both reviewers note the important gap in knowledge filled by your study and provide a number of suggestions for improvement of the manuscript. Please address all of these suggestions in the response letter with your resubmission.

Reviewer 1 suggests several additional experiments. Please note in your response if these experiments are added or why you feel they are not needed. Reviewer 2 has several suggestions for data presentation.

·

Basic reporting

ENGLISH LANGUAGE
Overall, the manuscript is clearly written, except for a few places were noted.


INTRODUCTION/REFERENCES
Line 46: While, the average age for diagnosis may be around 50, the average age for the cataract to affect vision is older. This statement may be misleading.
https://www.nei.nih.gov/learn-about-eye-health/resources-for-health-educators/eye-health-data-and-statistics/cataract-data-and-statistics
Line 54-55: The findings referenced in this paper by Serebryany need further explanation. The Serebryany paper is recent and a novel hypothesis.
Line 57: Provide reference supporting that gammaD is one of the most abundant lens proteins.
Line 66: Change misfold to unfold.
Line 68: An external factor is not required to cause unfolding. It could be aging. Please clarify.
Lines 85-87: The Forsythe et al 2019 reference should be grouped with the references on line 86 starting with Thorn 2019. The Pande reference in line 86 should be their paper from 2015, not 2010.
Lline 89: Define what is meant by non-equilibrium dynamics.
Line 95: Grammar needs to be corrected in this sentence- “Following studies have shed light into some features of the aggregation due to high concentration of metals”
Line 100- Change in-vivo to in vivo.
Line 101- Change cataracts lens to cataract lenses.
Line 108- Delete unique.

FIGURES:

Figure 1- Nice

Figure 2- Does the higher OD for gD with Zn mean more protein aggregated/precipitated? This could be confirmed with a SDS-PAGE gel.

Figure 3- Do all proteins undergo this increase in size with time? Maybe try BSA as a control? Does the unusual correlation function for gD with Cu mean anything? What do the shape of the curves mean? Also, panel C is not clear. How accurate is the instrument at the upper end of the sizes reported? Most importantly, can it be estimated from the DLS or turbidity measurements how much of the protein, ie., mass or concentration, is contributing to the increased light scattering?

Figure 4- How was the % unfolded calculated from the curves shown in Fig 4? It is stated that 10% of gD with Zn is unfolded. How much for with Cu? Perhaps best to also show the red shift in the peak F.I. Did it start at 330 nm and then shift to 335 nm? How much of a shift does 10% unfolded protein correspond to?

Figure 5- Is it correct that it could not be determined from the NMR where the Zn binds? Is this because the data also suggests aggregation in general?

Figure 6- Nicely presented and clear.

Supplementary Figures- unless there is a limit on the number of figures, supplementary figure 8 or 9 showing modeling of binding sites should be included as part of the main body of the text.

Experimental design

Original Primary research- Authors build on their previous work and that of several others of metal binding to gamma-crystallins. What is novel about this research is the more physiological conditions at which experiments were performed and the DLS, along with molecular modeling to fully extrapolate the meaning of the small changes.

Knowledge gap- Authors correctly state in the introduction that the starting process and pathway for protein aggregation is poorly understood.

Rigorous-
Line 131- Purity of expressed proteins should be confirmed not just with SDS-PAGE gels, but also with mass spectrometry if authors have access to this method.

Turbidity and DLS experiments should be done in triplicate if possible for statistical determination of a standard error of the mean of the radii. Given the error bars on the figures, it is possible then that each experiment had triplicates. Please state.

Experiments, especially the DLS experiments should be done at different concentrations, if possible.

Lines 526-527- How was the theoretical radius calculated for gD?

Methods sufficiently described- Yes. Thank you.

Validity of the findings

Robust data. Yes, the multiple methods to evaluate conformational changes and aggregation is excellent.

Limited speculation. Correct, except where stated in the discussion.

Conclusion is a nice succinct summary. Last line, Line 587, needs to be edited. However, the Discussion could be shortened, especially lines 440-495. The lines 497-518 are very intriguing and addresses Comment 4 below. This section is also speculative. Perhaps best to state clearly that it is speculative.

Additional comments

The most important findings of this paper are that unfolding is not required for aggregation as commonly thought and that metal binding can induce different pathways of aggregation, depending on the metal. These studies were performed at 37 C and therefore, are physiologically relevant and in contrast to previous studies. The conclusion that Zn induces a rearrangement, rather than unfolding and this is what causes aggregation is an exciting conclusion supported by the data.

Major comment 1- It is difficult to reconcile the small structural changes due to Zn, but yet large increase observed in radius in Figure 3. In contrast, gD with Cu does unfold in the NMR findings, but has a smaller radius in Figure 3. Please explain.

Major comment 2- Since only a very small % of aggregate can lead to a large increase in intensity of scattered light, is it possible to estimate the amount of protein in the sample that contributed to the light scattering? This would not be possible with DLS, but perhaps could be estimated from turbidity experiments if a gel was performed at the early time points of the soluble vs insoluble sample.

Major Comment 3- Authors' estimate that about 10% of gD with Cu is unfolded and is enough to cause aggregation. Is this correct? And, is it possible to estimate how much of the protein needs to be unfolded in order to scatter the light observed by turbidity and especially DLS? What % of the N-terminal domain residues have chemical shifts in the NMR? The unfolding occurs in the N-terminal domain, but all of the N-terminal domain?

Major Comment 4- While it is clear that gD with Cu unfolds and then aggregates, it is less clear how gD with Zn aggregates without unfolding. This is nicely discussed in the discussion. How will the aggregation pathway be determined for gD with Zn?

Reviewer 2 ·

Basic reporting

Description and Comments:
Understanding the mechanism of human gamma crystallin aggregation in age-related cataract is of paramount importance for the design of drugs to inhibit the process. In previous publications some of the authors of this ms. demonstrated that exposure of human recombinant gamma D crystallin to Zn2+ or Cu2+ induced protein aggregation. However, the precise mechanism of aggregation was unclear, especially since Cu2+ is redox active while Zn2+ which is inert.
In the study, the first determined the lowest stoichiometric Mn2+ : protein concentration able to induce protein aggregation and found two distinct aggregation patterns based on turbidimetry. Zn2+ induced immediate aggregation while Cu2+ induced aggregation after a lag phase. To further characterize the aggregation mechanism and the respective role of each protein residue they used a combination of DLS, fluorescence and NMR techniques. In a nutshell, they propose that Zn2+ promotes protein aggregation by forming metal bridges involving binding to X and Y residues in the N-terminal domain and Y residues in the C-terminal domain. In contrast their data suggest that Cu2+ promotes aggregation first via unfolding of the N-terminal domains in a mechanism involving binding to His X in the N-terminal domain.
This is a very well written and intelligently conducted study that involves a large amount of high quality data that convincingly help solve a problem of critical importance for our understanding of the role of bivalent metals in gamma crystallin aggregation. As such this reviewer has only minor suggestions for improvement:
1) The DLS data would benefit from being presented in table format. The reviewer had to bounce back and forth between the text and the figures to grab the overview and understand how the DLS parameters were being affected by presence of metal and incubation time.
2) NMR studies and residue specific changes: Here again, a table summarizing the findings would make it easier to the reader to grab the essential changes.
3) I was able to grab the proposed binding sites of the metals, but had difficulties seeing what parts of the protein is being unfolded. Can the authors provide a blow-up picture of the native and unfolding N-terminal domain?
4) Line 514-518: If spacewise feasible, such results should be mentioned I the Abstract, Similarly for the Cu(II) results.
5) Reversibility of aggregation. You mention Zn2+ induced aggregation is reversible. How about Cu2+? Can the aggregation be stopped beyond the lag phase?
6) While the English language is excellent, there a few typographical mistakes that need fixing. E.g. Fig. S2 legend “describe” instead of “described”. Line 252; “get insight into” rather than “on”
7) Fig. S7 legend: state “induced by Zn(II) rather than “metal ion”.
In summary, no new experiments are needed. However, since there presumably is not enough space in the Abstract to present specific findings, summary tables are needed in the Main Text to help the reader remember key findings.

Experimental design

Very strong throughout ! See above for other comments

Validity of the findings

Robust and sound as far as I can tell.

Additional comments

You are to be commended for a superb study. The results and your conclusions appear to be fully warranted. I am not an expert in protein-NMR and metal binding, thus experts in this field might have various recommendations

---

## Round 0.2 · accepted · Accept

Thank you for addressing the reviewer comments and for your revised submission. I am happy to now accept your manuscript for publication in PeerJ.

Both reviewers had some minor additional comments that I encourage you to include in your final version for publication.

You will be given the option to make the reviews of your manuscript available to readers. Please consider doing so as this review record can be a great resource for readers of your paper and contributes to more transparent science.

Thank you again for your contribution.

·

Basic reporting

Please include the following reference, which would have been available before this manuscript was submitted. While the protein is different, gammaS-crystallin, it has homology to gammaD-crystallin in this paper and there is some overlap of experiments and the author's findings should be discussed in light of the findings for gammaS-crystallin.

Divalent Cations and the Divergence of βγ-Crystallin Function.
Roskamp KW, Kozlyuk N, Sengupta S, Bierma JC, Martin RW.
Biochemistry. 2019 Nov 12;58(45):4505-4518. doi: 10.1021/acs.biochem.9b00507. Epub 2019 Nov 1.
PMID: 31647219

Additionally, while gammaD-crystallin is one of the gamma-crystallins in the very young lens, of the 3 gamma-crystallins-C, D, and S, it is found in the least amount. In the aged adult lens, there is very little gammaD-crystallin left compared to gammaS-crystallin. The current reference in the introduction supporting your statement that gammaD-crystallin is one of the major gamma-crystallins in the nucleus is not correct. Please include the appropriate reference and revise the statement to reflect the relevance of gammaD-crystallin in the whole lens.

Quantitative measurement of young human eye lens crystallins by direct injection Fourier transform ion cyclotron resonance mass spectrometry.
Robinson NE, Lampi KJ, Speir JP, Kruppa G, Easterling M, Robinson AB.
Mol Vis. 2006 Jun 21;12:704-11.
PMID: 16807530

Experimental design

Fine.

Validity of the findings

Fine.

Additional comments

See above references to include.

Reviewer 2 ·

Basic reporting

Novel and exciting, well written study with important findings that will help us understand the potential role of bivalent metals in age-related cataract

Experimental design

very strong and rigorous study with meticulous attention to details

Validity of the findings

strong and rigorous

Additional comments

Congratulations for a very nice paper and revisions!
Addition of Table 2 and 3 are helpful.
Table 3, legend:
What do you mean by the “remains residues were not assigned or overlapped”.
Do you mean the “remaining residues were not assigned or were not overlapping"